# Noise Analysis of a Passive Resonant Laser Gyroscope

**DOI:** 10.3390/s20185369

**Published:** 2020-09-19

**Authors:** Kui Liu, Fenglei Zhang, Zongyang Li, Xiaohua Feng, Ke Li, Yuanbo Du, Karl Ulrich Schreiber, Zehuang Lu, Jie Zhang

**Affiliations:** 1MOE Key Laboratory of Fundamental Physical Quantities Measurements & Hubei Key Laboratory of Gravitation and Quantum Physics, PGMF and School of Physics, Huazhong University of Science and Technology, Wuhan 430074, China; kuiliu@hust.edu.cn (K.L.); fenglei_zhang@hust.edu.cn (F.Z.); zongyang_li@hust.edu.cn (Z.L.); xiaohua_feng@hust.edu.cn (X.F.); ke_li@hust.edu.cn (K.L.); yuanbodu@hust.edu.cn (Y.D.); zehuanglu@mail.hust.edu.cn (Z.L.); 2Technical University of Munich, Forschungseinrichtung Satellitengeodäsie, Geodetic Observatory Wettzell, 93444 Bad Kötzting, Germany; ulrich.schreiber@tum.de

**Keywords:** passive resonant laser gyroscope, Earth rotation, Sagnac effect, noise analysis

## Abstract

Large-scale laser gyroscopes have found important applications in Earth sciences due to their self-sufficient property of measurement of the Earth’s rotation without any external references. In order to extend the relative rotation measurement accuracy to a better level so that it can be used for the determination of the Earth orientation parameters (EOP), we investigate the limitations in a passive resonant laser gyroscope (PRG) developed at Huazhong University of Science and Technology (HUST) to pave the way for future development. We identify the noise sources from the derived noise transfer function of the PRG. In the frequency range below $10^{-2}\;\mathrm{Hz}$, the contribution of free-spectral-range (FSR) variation is the dominant limitation, which comes from the drift of the ring cavity length. In the $10^{-2}$ to $10^{3}\;\mathrm{Hz}$ frequency range, the limitation is due to the noises of the frequency discrimination system, which mainly comes from the residual amplitude modulation (RAM) in the frequency range below 2 Hz. In addition, the noise contributed by the Mach–Zehnder-type beam combiner is also noticeable in the 0.01 to 2 Hz frequency range. Finally, possible schemes for future improvement are also discussed.

## 1. Introduction

Optical interferometers have important applications in the field of precision measurement. Large-scale interferometers have much better sensitivity and resolution; therefore, many ground-breaking works benefiting from their high sensitivities have been reported. For example, large-scale Michelson interferometers, such as LIGO and VIRGO, make it possible to detect extremely weak gravitational wave signals [1,2,3]. In the meantime, large-scale Sagnac interferometers play a vital role in measuring the rotation of the Earth [4].

The Sagnac effect is useful in the detention of rotational signals. Instruments based on the Sagnac effect have many applications in different fields [4,5,6,7,8,9,10,11,12,13,14,15,16]. In particular, large-scale optical gyroscopes find applications in inertial navigation, geophysical study, seismic isolation, platform stabilization, etc. [4,6,7,8,9]. Optical gyroscopes utilize the non-reciprocal phenomenon inside a ring cavity introduced by the rotation of the cavity frame based on the Sagnac effect, which means two beams would experience an unequal round-trip travel time in opposite directions inside an identical light path of a rotating ring interferometer. The interferometric gyroscopes convert the travel time difference into the accumulated phase difference of the opposite beams, while the resonant gyroscopes measure the frequency difference by utilizing a ring cavity instead. The state-of-the-art interferometric fiber optical gyroscopes can reach a sensitivity of $10^{-9}\;\mathrm{rad}/s/\sqrt{\mathrm{Hz}}$ [10,11]. Recently, micro-optical gyroscopes made of whispering gallery mode resonators have attracted much attention and are believed to have great potential to be applied to both industrial and navigational fields [12,13]. On the other hand, a branch of resonant laser gyroscopes has evolved from compact aircraft inertial sensors back into large-scale and complex laboratory setups for applications in the geosciences and fundamental physics in recent decades, with sensitivity as high as $10^{-11}\;\mathrm{rad}/s/\sqrt{\mathrm{Hz}}$ [4,7,14,15,16].

The detection signal for a horizontally placed geodetic laser gyroscope at some co-latitude θ on the Earth is proportional to the rotation rate and can be expressed as [4]:(1)$$f_{s}=S\cdot \vec{n}\cdot \vec{\Omega }=K_{s}\Omega =\frac{4A}{\lambda P}\Omega \cos\theta ,$$
where $f_{s}$ is the output frequency of the sensor (Sagnac frequency), S=4A/λP is the geometrical scale factor, $K_{s}=\frac{4A}{\lambda P}\cos\theta$ is the orientated scale factor, *A* is the area enclosed by the light path of the gyroscope, *P* is the perimeter of the ring cavity, λ is the laser wavelength, Ω is the Earth rotation rate, and θ is the north-south angle of projection. The cosθ term indicates that the sensor is only sensitive to the projection of the vector of rotation to the nominal normal direction of the cavity enclosed area. The sensitivity and resolution of the sensor are proportional to its dimensions. Large-scale laser gyroscopes have found important applications in the geosciences due to their extremely high sensitivity.

One challenging application of these large-scale sensors is in the field of fundamental physics research. For example, detecting the Lense–Thirring effect in a terrestrial laboratory is one of the ambitious goals [14]. The Lense–Thirring effect induces a very small DC signal to the Earth rotation vector, with a magnitude of a $10^{-10}$ Earth rotation rate ($10^{-10}\Omega_{E}$) [4,17,18]. Therefore, in order to perform this test, the relative Earth rotation rate measurement accuracy has to reach the level of $10^{-10}$. More progress is required even if one can relax the strict requirements by means of a multi-axis large-scale gyroscope design [17]. The establishment of the absolute accuracy of the sensor should be given a high priority [19].

Ring laser technology has kick-started the field of rotational seismology, where ultimately a rotation measurement resolution of $10^{-14}$−1 rad/s and a frequency bandwidth of 3mHz–50Hz are demanded [4,20]. Large-scale laser gyroscopes are the only ground rotation sensors that meet these demands of high resolution, wide dynamic range, and broad bandwidth at the same time. Because these instruments are only sensitive to rotational motions, additional information on ground motions can be obtained through this extra-dimensional metrology. They are opening a new window for rotational seismology by filling the gap of lacking high-performance rotation detection instruments [21].

Our aim is to apply it to the field of geodesy. The celestial and the terrestrial reference frames are tied together by the exact knowledge of Earth’s orientation and Earth’s rotation. Inertial sensing gyroscopes are directly linked to the Earth rotation vector as shown in Equation (Equation 1). The large ring laser “G” located at the Geodetic Observatory Wettzell started to record the polar motion in 2010, and the results are consistent with the data provided by the International Earth Rotation and Reference Systems Service (IERS), which is obtained from the space geodetic techniques [4,22,23]. The full Earth rotation vector is already provided by a ring laser array and it is also getting close to the detection of the length of day (LoD) fluctuations by using this type of inertial sensing technique [24,25]. Recording the universal time (UT1) data in hourly temporal resolution would be a major supplement to the space geodetic observations that are used to determine the Earth orientation parameters (EOP) [26].

One thing that is common for these applications is the demand for the high sensitivity of the gyroscopes and long-term stability. The large-scale laser gyroscope has proven to be an effective solution to push the sensitivity forward by increasing the size of the sensor; however, a larger scale factor may easily compromise the stability. There are two types of large-scale laser gyroscopes: ring laser gyroscopes (RLGs) and passive resonant laser gyroscopes (PRGs). Although the techniques are slightly different, they both share the same significant potential to be applied to the Earth sciences. The large-scale RLGs have a long history and have gone through a series of systematic investigations. Therefore, the technical noise sources and the limitations through the systematic biases of the RLGs are well understood, such as back-scatter coupling, scale factor fluctuation, orientation fluctuations, etc. The fundamental noise floor of the RLGs is shot-noise, which comes from the spontaneous emission of the gain medium and is inversely proportional to the square root of the extracted laser power. For the large-scale RLGs, the free-spectral-ranges (FSRs) are too small in comparison to the gain bandwidth of He-Ne, so to avoid mode competitions inside the RLGs and to keep the instruments running stably, the single longitudinal mode laser output power is usually limited to some nano-watts, which increases the shot-noise levels [4].

The fundamental noise of the PRGs is similar, but is limited by the injection laser power instead, which indicates that the PRGs potentially can have a lower fundamental shot-noise floor with higher laser power [27]. Successful examples have been carried out in gravitational wave detection by using a passive cavity-enhanced sensing technology [1]. Another advantage of the PRG over the RLG is that it deals with an empty optical ring cavity, rather than an optical cavity filled with a gain medium, so in the absence of non-linear laser dynamics, it is easier to estimate the absolute accuracy in EOP measurements. However, the performance of the PRGs is still below that of the RLGs at this time, which suggests that there are still other obstacles to be overcome in the development of the PRGs [7,15]. Therefore, it is desirable to take a closer look at these instruments and to search for the limitations.

In this paper, we make a full investigation of the noise sources in a PRG with a size of 1m×1m developed at Huazhong University of Science and Technology (HUST) [15]. We find that the sensor noise in a PRG is dominated by the FSR drift and by laser frequency locking noise. Other noise sources, such as phase noise in the frequency beating measurement and frequency noise induced by the diode laser and frequency shifters are also taken into consideration. In Section 2, we review the experimental setup of our PRG. In Section 3, we derive the noise transfer function from a typical feedback control model and identify the noise sources. We show that the noise may come from the free-running laser, the frequency shifter, the frequency discrimination system, the cavity length fluctuation, and the servo amplifier. In Section 4, we introduce the quantitative analysis of noise sources in the PRG, investigating their contributions and showing that the contributions of the FSR and the frequency discrimination system are the current dominant limitations. We further show that the scale factor fluctuation is not a problem because of the self-compensation mechanism. In Section 5, we summarize and point out directions to further improve the PRG technology.

## 2. Experimental Setup

The experimental setup is shown in Figure 1 [15]. The central part of our PRG is a 1m×1m square ring cavity, which is composed out of four super mirrors, which provide a quality factor of $5.3\times 10^{11}$. The operating wavelength is 1064 nm. To increase the dimensional stability of the cavity, the four mirrors are held in customized mirror mounts, which are fixed to a base plate that is made of granite with a size of 1.8m×1.8m×0.25m. The whole ring cavity including the light paths is enclosed in a $10^{-6}$ Pa-level vacuum system that is rigidly anchored on the granite to avoid disturbances from air flow. At each corner of the vacuum system, there are windows to allow laser beams to enter and exit. The PRG sits on top of six steel legs and is located in a cave laboratory, which has low seismic noise and small temperature fluctuations.

A 1064 nm diode laser is utilized as the injected light source. The output of the laser is split in two and is injected into the clockwise (CW) and counter-clockwise (CCW) directions simultaneously. We used the Pound–Drever–Hall (PDH) method to lock the beams to the cavity [28]. In the two injection light paths, we used electro-optic modulators to produce phase-modulated laser beams. The phase-modulated laser beam reflected from the coupling mirror allows the detection of the frequency detuning between the laser and the cavity resonance after the demodulation process. In order to obtain a tight locking of the laser to the ring cavity in the CCW direction, we implemented a two-branch feedback control loop of the diode laser frequency. The fast feedback branch is set by controlling the driving current of the diode laser, while the slow feedback branch uses a piezo actuator inside the laser head.

For the CW direction, we utilized an acousto-optic modulator (AOM) as a frequency shifter to compensate the frequency detuning between the injected beam and the cavity resonance. The two locking loops in the CCW and CW directions are named as the primary and the secondary loop, respectively. There are two beams oscillating in the ring cavity after the two locking-loops are closed, with each beam containing a small fraction of the amplitude of the other beam from the back-scatter coupling, resulting in beating signals that can be observed on each of the output beams. In our case, the driving frequency of the AOM is approximately 75 MHz to match one FSR of the square ring cavity. In this way, the observed beating signals caused by back-scattered light can be shifted to 75 MHz, which can be easily separated from the desired PDH signal in the frequency domain. Thus, the locking perturbations caused by the back-scattered light can be significantly reduced. Behind the ring cavity, the two leak-out beams are superimposed by a Mach–Zehnder-type beam combiner. The frequency difference of these two beams is the sum of the Sagnac frequency and the FSR and is detected by an avalanche photo-diode (APD).

## 3. Control Model

We used the PDH locking method to realize the frequency locking of the two injected laser beams to the resonant peaks of the ring cavity in the PRG. Since the PDH system is a closed-loop locking system, it is necessary to model the entire set of feedback loops and use the noise transfer function to identify the noise sources to investigate their contributions. With that, we can understand the major limitations of our PRG and find the indications for further improvement.

From the perspective of control theory, a typical block diagram of the feedback control loop in a laser frequency stabilization system was given by T. Day et al. and shown in Figure 2 [29,30]. We denote the noise contribution of the laser source as $S_{\nu ,laser}$, the noise contribution of the frequency discrimination system as $S_{\nu ,disc}$, and the noise contribution of the servo feedback system as $S_{\nu ,servo}$. It should be noted that the noise contribution of the actuator is indistinguishable from the noise of the laser itself. Therefore, it is not listed separately here.

The error signal is obtained through the frequency discrimination system, where a conversion coefficient called the frequency discrimination slope is denoted as *D* in units of V/Hz. The conversion gain of the servo amplifier is denoted as *G* in units of V/V. The output of the servo amplifier acts on the actuator of the laser to correct the frequency detuning between the laser and the ring cavity resonance. The conversion factor of the actuator is denoted as *K* in units of Hz/V. In the closed-loop condition, the total closed-loop frequency noise power spectral density can be expressed as [29,30]:(2)$$S_{\nu ,cl}^{2}=\frac{S_{\nu ,laser}^{2}+K^{2}S_{\nu ,servo}^{2}+K^{2}G^{2}S_{\nu ,disc}^{2}}{{\left|1+KGD\right|}^{2}},$$
where the units of $S_{\nu ,cl}^{2}$ are ${\mathrm{Hz}}^{2}/\mathrm{Hz}$. It can be seen from Equation (Equation 2) that the closed-loop frequency noise depends on the open-loop gain of the system. For a reliable locking, the condition of KGD≫1 should be met. Thus, Equation (Equation 2) can be simplified to:(3)$$S_{\nu ,cl}\approx \frac{S_{\nu ,disc}}{D}.$$

Equation (Equation 3) shows that the closed-loop frequency noise is dominated by the noise of the frequency discrimination system, which includes laser shot-noise, the electronic noise of the photo-detector, the frequency noise of the modulation signal source, the 1/f noise of the electronic amplifier, and the noise from residual amplitude modulation (RAM). Obviously, the larger the frequency discrimination slope *D* is, the smaller the closed-loop frequency noise is.

For our PRG, there are two locking loops to lock the injection laser frequencies to the same cavity in the CCW and CW directions, respectively; thus, the two laser beams see the same fluctuation of the cavity. In addition, the two laser beams are from the same source, and most of the laser frequency noise cancels out as well. Therefore, in an ideal case, the contribution of the laser frequency noise and the cavity length fluctuation noise to the gyroscope output would be significantly reduced due to the common-mode rejection mechanism. In reality, we should take these noise sources into account because of the finite common-mode rejection ratio. The block diagram of the control model in our PRG is depicted in Figure 3, where a second loop is introduced. It consists of two PDH control loops, where the CCW loop (primary loop) locks the laser frequency to the ring cavity at a frequency of $\nu_{out,1}$. Suppose the free-running laser frequency is $\nu_{fr}$, then the closed-loop laser frequency of the primary loop is given by:(4)$$\nu_{out,1}=\nu_{fr}-K_{1}\left\{G_{1}\left[D_{1}\left(\nu_{out,1}-\nu_{ccw}\right)+e_{disc,1}\right]+e_{servo,1}\right\}+f_{AOM1},$$
where $K_{1}$ and $D_{1}$ are the conversion coefficient of the actuator and the discriminator slope in the primary loop, $\nu_{ccw}$ is the resonant eigen-frequency of the ring cavity in the CCW direction, $f_{AOM1}$ is the driving frequency of AOM1, and $e_{disc,1}$ and $e_{servo,1}$ are the electronic noises in the discriminator and the servo amplifier of the primary loop. Equation (Equation 4) can be expressed as:(5)$$\nu_{out,1}-\nu_{ccw}=\frac{\left(\nu_{fr}+f_{AOM1}-\nu_{ccw}\right)-K_{1}G_{1}e_{disc,1}-K_{1}e_{servo,1}}{1+K_{1}G_{1}D_{1}}.$$

Similarly, we derive the closed-loop laser frequency of the secondary loop as:(6)$$\nu_{out,2}-\nu_{cw}=\frac{\left(\nu_{out,1}-f_{AOM1}+f_{AOM2}-\nu_{cw}\right)-K_{2}G_{2}e_{disc,2}-K_{2}e_{servo,2}}{1+K_{2}G_{2}D_{2}},$$
where $\nu_{out,2}$ is the laser frequency locked to the cavity in the CW direction, $K_{2}$ and $D_{2}$ the conversion coefficient of the actuator and the discriminator slope in the secondary loop, $\nu_{cw}$ the resonant eigen-frequency of the ring cavity in the CW direction, $f_{AOM2}$ the driving frequency of AOM2, and $e_{disc,2}$ and $e_{servo,2}$ the electronic noises in the discriminator and the servo amplifier of the secondary loop. We subtract Equation (Equation 5) from Equation (Equation 6) to obtain the expression of the frequency difference between the two laser beams:(7)$$\begin{array}{cc} \nu_{out,2}-\nu_{out,1}=\nu_{cw}-\nu_{ccw} & +\frac{\left(\nu_{out,1}-f_{AOM1}+f_{AOM2}-\nu_{cw}\right)-K_{2}G_{2}e_{disc,2}-K_{2}e_{servo,2}}{1+K_{2}G_{2}D_{2}}\\ \; & -\frac{\left(\nu_{fr}+f_{AOM1}-\nu_{ccw}\right)-K_{1}G_{1}e_{disc,1}-K_{1}e_{servo,1}}{1+K_{1}G_{1}D_{1}}. \end{array}$$

Note that $\nu_{out,2}-\nu_{out,1}$ represents the beat note of the two laser beams after locking, and $\nu_{cw}-\nu_{ccw}$ is the resonant eigen-frequency difference in the CW and CCW directions and is equal to the Sagnac frequency. Using Equation (Equation 7), we can obtain the noise in the beat note:(8)$$\begin{array}{cc} S_{\nu_{2}-\nu_{1}}^{2}\approx \frac{K_{2}^{2}G_{2}^{2}D_{2}^{2}S_{\nu ,sagnac}^{2}}{{\left|1+K_{2}G_{2}D_{2}\right|}^{2}} & +\frac{S_{\nu ,AOM1}^{2}+S_{\nu ,AOM2}^{2}}{{\left|K_{2}G_{2}D_{2}\right|}^{2}}+\frac{S_{\nu ,AOM1}^{2}+S_{laser,1}^{2}+S_{\nu ,cavity}^{2}}{{\left|K_{1}G_{1}D_{1}\right|}^{2}}\\ \; & +\frac{S_{disc,2}^{2}}{D_{2}^{2}}+\frac{S_{disc,1}^{2}}{D_{1}^{2}}+\frac{S_{servo,2}^{2}}{G_{2}^{2}D_{2}^{2}}+\frac{S_{servo,1}^{2}}{G_{1}^{2}D_{1}^{2}}, \end{array}$$
where $S_{\nu_{2}-\nu_{1}}^{2}$ is the power spectral density of the beat note in units of ${\mathrm{Hz}}^{2}/\mathrm{Hz}$. The first term $S_{\nu ,sagnac}^{2}$ stands for the noise power spectral density of the Sagnac frequency. The noise of a free-running laser is $S_{laser,1}^{2}$; the noise introduced by the AOM(*i*) is $S_{\nu ,AOM(i)}^{2}$; the noise of the frequency discrimination system is $S_{disc,(i)}^{2}$; the equivalent frequency noise caused by cavity fluctuation is $S_{\nu ,cavity}^{2}$; and the noise of the servo amplifier is $S_{servo,(i)}^{2}$. We also use the approximation of $1+K_{i}G_{i}D_{i}\approx K_{i}G_{i}D_{i}$ except for the first term, which is essential to clarify the importance of high loop gain for real Sagnac frequency measurement.

It is worth noting that Equation (Equation 8) is only suitable for a common-mode operation of a PRG, which means that the two beams are locked to the same longitudinal mode. In our setup, the PRG is running on two different longitudinal mode indices. When an optical cavity is excited by an external laser source, the cavity houses an integer number *m* of wavelengths, such that P=mλ, where *m* is the longitudinal mode index and satisfies $m=\nu_{laser}/FSR$. Then, the noise in the beat note of the two oscillating beams can be modified as:(9)$$\begin{array}{cc} S_{\nu_{2}-\nu_{1}}^{2}\approx \frac{K_{2}^{2}G_{2}^{2}D_{2}^{2}S_{\nu ,sagnac}^{2}}{{\left|1+K_{2}G_{2}D_{2}\right|}^{2}} & +\frac{S_{\nu ,AOM2}^{2}}{{\left|K_{2}G_{2}D_{2}\right|}^{2}}+\frac{S_{laser,1}^{2}+S_{\nu ,cavity}^{2}}{{\left|K_{1}G_{1}D_{1}\right|}^{2}}+\frac{S_{\nu ,cavity}^{2}}{m^{2}}\\ \; & +\frac{S_{disc,2}^{2}}{D_{2}^{2}}+\frac{S_{disc,1}^{2}}{D_{1}^{2}}+\frac{S_{servo,2}^{2}}{G_{2}^{2}D_{2}^{2}}+\frac{S_{servo,1}^{2}}{G_{1}^{2}D_{1}^{2}}. \end{array}$$

## 4. Noise Analysis

Using the model in control theory, we derive the noise transfer function in a PRG, as shown in Equations (Equation 8) and (Equation 9). We identify the noise sources from the noise transfer function as: discriminator noise, cavity length fluctuation noise, laser frequency noise, and the servo amplifier noise. However, there are other noise sources in the first term $S_{\nu ,sagnac}^{2}$ that are not clear from Equations (Equation 8) and (Equation 9). We summarize the noise sources in $S_{\nu ,sagnac}^{2}$ as: beam combiner noise and the scale factor fluctuation noise, which are partly evident in Equation (Equation 1). In the following, we discuss the quantitative analysis of noise sources in the PRG.

To establish the contribution from every part in the noise expression, we have to firstly measure the frequency discrimination slope $D_{i}$, the conversion gain of the servo amplifier $G_{i}$, and the conversion factor of the actuator $K_{i}$. The frequency discrimination slope $D_{i}$ is the coefficient that converts the frequency detuning between the laser and the cavity resonant peak into voltage. Actually, $D_{i}$ has a dependence on frequency ω and can be given by [30]:(10)$$\begin{array}{c} D\left(\omega \right)=\frac{D_{0}}{1+2j\omega /(2\pi \nu_{c})}, \end{array}$$
where $\nu_{c}$ is the linewidth of the ring cavity and $D_{0}$ is the flat response coefficient within the cavity linewidth. We can obtain the cavity linewidth through a cavity ring-down time measurement. To measure $D_{0}$, we use a modulation scheme. As shown in Figure 4a, we add a modulation signal to the error signal in the secondary loop, while the two feedback loops maintain locking. Thus, there is a corresponding signal in response to the laser frequency of the secondary loop. We can use an APD behind the cavity to observe it by the beating signal of the two beams, where the beam of the primary loop is set as the reference. If the amplitude of the modulation signal $V_{m}$ is small, then the laser frequency is completely within the cavity linewidth during the entire process, which can be considered as a linear response. We use different amplitudes of square wave modulation signals and record the corresponding responses as shown in Figure 4b. The coefficient $D_{0}$ then can be obtained by a linear fitting, as shown in Figure 4c. To measure the coefficient $D_{0}$ in the primary loop, we only need to switch over the CCW and CW loop. Here, the value of $D_{0}=1.28\pm 0.02\;\mathrm{mV}/\mathrm{Hz}$ is identical for both loops within experimental uncertainty.

We obtain the gain of the servo amplifier $G_{i}$ with the help of a network analyzer. In order to measure the conversion factor of the actuator *K*, a modulation method similar as the one used in the measurement of $D_{i}$ is implemented. When a modulation signal is fed to the laser driver, we record the frequency response of the laser by beating the diode laser with an ultra-stable reference laser [31,32]. Since there are two actuators inside the laser head, we measure the coefficients independently for the current and PZT. The measured conversion factor of the laser current is $K_{c}\;=\;\left(1.510\pm 0.001\right)\;\times 10^{8}\;\mathrm{Hz}/V$, and the conversion coefficient of the PZT is $K_{p}=\left(3.997\pm 0.001\right)\times 10^{9}\;\mathrm{Hz}/V$. The actuator in the secondary loop is an AOM driving signal generator, and the measured conversion coefficient is $K_{2}=\left(1.955\pm 0.001\right)\times 10^{4}\;\mathrm{Hz}/V$. The total loop gain of the primary and secondary loop is summarized in Figure 5, where the predicted curves are calculated by the poles-zeros settings of the proportional-integral-derivative (PID) regulators. The servo regulator behaves as a limited integrator when the current feedback is activated in the primary loop only and behaves as an unlimited integrator when the PZT feedback loop is added as well. The loop gain is about 118 dB and 169 dB at 1 Hz before and after the double-branch feedback loop is switched on. As for the secondary loop, it has only one branch and behaves as a limited integrator with a gain of 55 dB in the frequency band below 1 Hz. The predicted gain values are used to calculate the suppression ratios of the cavity length fluctuation and the laser frequency noise.

We obtain the rotation detection performance of the PRG by analyzing the beat note detected by the APD behind the ring cavity, which is depicted in Figure 6, colored in blue. In the following, we describe the noise contributions in the system in detail, such as the discriminator noise, the cavity length fluctuation, the laser frequency noise, the beam combiner noise, and the scale factor fluctuation, which are mentioned in Section 3.

### 4.1. Discriminator Noise

The discriminator noise $S_{disc,(i)}^{2}$ is dominated by the shot-noise, the electronic noise of the photo-detector, and the noise from the RAM. We can measure $S_{disc,(i)}^{2}$ in an off-line mode at the demodulation terminal, where the status of the instrument remains the same as it is in the closed-loop, except for the case when the locking loops are open. As already mentioned, the contribution of the discriminator noise has a frequency dependent discrimination slope $D_{i}$, which is mainly determined by the Q-factor of the cavity itself. The parameter $D_{i}$ is already measured; therefore, we can obtain the total contribution of the discriminator noise, which is depicted in Figure 6, colored in red.

In the PDH locking systems, the modulation frequencies are set to 22 MHz and 28 MHz. The amplitude noise of the laser in this frequency band can be ignored, and we should take the shot-noise into account first. In the experiment, the incident light power $P_{0}$ is 110.0 µW, and the calculated shot-noise contribution $\kappa R\sqrt{2h\nu P_{0}}$ is about $3.4\times 10^{-7}\;V/\sqrt{\mathrm{Hz}}$, taking into account the conversion loss of the frequency mixer κ and the PD responsivity *R* [30]. Using the discrimination slope parameter, the shot-noise contribution to the rotation sensitivity is $6.0\times 10^{-10}\;\mathrm{rad}/s/\sqrt{\mathrm{Hz}}$, which is the shot-noise limit of our PRG and is shown with the red dashed line in Figure 6. However, the detection power of the laser beams is not high enough. For this reason, the electronic noise of the frequency discrimination system cannot be neglected. We can directly measure the total noise in the discrimination system contribution at the demodulation terminal, which is $\left(8.0\pm 0.4\right)\times 10^{-7}\;V/\sqrt{\mathrm{Hz}}$. This indicates that the electronic noise is dominant in the frequency discrimination system at a frequency range above 1 Hz. The noise introduced by the RAM is quite common in a PDH system [33]. The RAM effect will ruin the stability of the locking zero-baseline, thus affecting the stability of the Sagnac frequency and ultimately the performance of our PRG. The red solid curve in Figure 6 shows that the noise caused by the RAM effect acts like a 1/f noise between 0.01 and 1 Hz. It is clear that these noise sources in the discrimination system are the dominant limitations in the PRG. This also suggests that in order to push the PRG to a better performance, the attendance to the noise sources in the discrimination system has the highest priority.

### 4.2. Cavity Length Fluctuation

During the operation on different longitudinal mode indices, the cavity length fluctuations appear in two types of noise contributions. The first one is the residual cavity length fluctuation noise due to the finite primary loop gain. Assuming the loop gain is infinite over the entire frequency range, then the laser frequency would perfectly follow the cavity fluctuations, once the primary servo loop is closed. The secondary loop is only required to compensate the detuning between the laser frequency and the cavity resonance in the CW direction introduced by the rotation of the Earth. Since the laser beams used for the two servo loops are from one single diode laser, the cavity length fluctuation is a common-mode noise and can be suppressed by the primary loop gain. In reality, the primary loop gain cannot be infinite. Therefore, we express the the residual cavity length fluctuation noise as $S_{\nu ,cavity}^{2}/\left({\left|K_{1}G_{1}D_{1}\right|}^{2}\right)$ in units of ${\mathrm{Hz}}^{2}/\mathrm{Hz}$, where $S_{\nu ,cavity}^{2}$ represents the equivalent frequency noise introduced by the cavity length fluctuation.

The second contribution is caused by the change of the FSR [15], which depends on the longitudinal mode index *m* and can be expressed as $S_{\nu ,cavity}^{2}/m^{2}$. We obtain the cavity length fluctuation noise by beating the gyro laser with the ultra-stable laser as a reference when the primary servo loop is closed [31]. Since the noise of the ultra-stable laser is negligible, the beat signal fluctuations represent the variation in cavity length. We measure the primary loop gain $K_{1}G_{1}D_{1}$, and the longitudinal mode index *m* can be calculated as $m=\left(3.7528\pm 0.0002\right)\times 10^{6}$, then the contributions of these two types are both obtained, which are colored in pink and black in Figure 6, respectively. It can be seen that the noise contributions of these two items in the frequency band above $10^{-2}\;\mathrm{Hz}$ are not the main limitations. However, in the low-frequency range below $10^{-2}\;\mathrm{Hz}$, the contribution of the FSR change is in good agreement with the noise in the detected beat frequency. This is because the actual detected beat frequency contains the composition of one FSR. Therefore, it is very important to remove the contribution of the FSR. The residual cavity length fluctuation noise is not a limitation at this moment, which benefits from the sufficient primary loop gain shown in Figure 5a. If we implement an active stabilization scheme, like in [16,32,34], both of them can be suppressed significantly.

### 4.3. Laser Frequency Noise

There are two sources of laser frequency noise in our PRG: the first one is the free-running laser frequency noise, and the other is the additional frequency noise introduced by the frequency-shifting device. The free-running laser frequency noise is similar to the noise introduced by the cavity length fluctuations and has a dependence on the primary loop gain $K_{1}G_{1}D_{1}$. The residual frequency noise of the free-running laser after the common-mode rejection can be given by $S_{laser,1}^{2}/\left({\left|K_{1}G_{1}D_{1}\right|}^{2}\right)$. In the secondary loop, there is a frequency shifter AOM2, which introduces frequency noise through the driving signal source. The noise from AOM2 is suppressed by the ratio of the loop gain $K_{2}G_{2}D_{2}$ of the CW loop, which can be expressed as $S_{\nu ,AOM2}^{2}/\left({\left|K_{2}G_{2}D_{2}\right|}^{2}\right)$. We can obtain the free-running laser frequency noise by measuring the beat frequency of the free-running laser and the ultra-stable reference laser. The frequency noise above 2 Hz is measured with a phase noise analyzer, and the frequency noise below 2 Hz is obtained from a frequency counter. The frequency noise of the driving signal source of AOM2 can be directly measured by the phase noise analyzer and the frequency counter. The obtained results are converted to the equivalent rotational noise as shown in Figure 6. The orange curve is the contribution of the residual frequency noise of the laser, and the green curve is the contribution of the additional laser frequency noise introduced by AOM2. It can be found that they are not the limitations affecting the performance of the PRG, though they are above the shot-noise limit in some parts of the frequency band. Since the loop gain of the secondary loop is only 55 dB in the frequency band below 1 Hz, as shown in Figure 5b, a broader bandwidth of the locking loop and higher loop gains are required to suppress the noise contributions further, which is essential to reach the shot-noise of this instrument.

### 4.4. Beam Combiner Noise

The Sagnac frequency is obtained at one of the corners of the ring cavity, where a Mach–Zehnder type beam combiner is used to make the two leak-out beams superimposed and an APD is used to detect the beat signal. The noise sources here include shot-noise from the APD and environmental disturbances in the beam path. The phase fluctuation caused by the shot-noise on the APD is given by:(11)$$S_{\phi }=\sqrt{\frac{h\nu_{0}}{P_{tr}}},$$
then the frequency noise is:(12)$$S_{f_{tr}}=fS_{\phi }=f\sqrt{\frac{h\nu_{0}}{P_{tr}}},$$
where $S_{f_{tr}}$ is given in units of $\mathrm{Hz}/\sqrt{\mathrm{Hz}}$, *f* is the Fourier frequency, *h* is the Planck constant, $\nu_{0}$ is the laser frequency, and $P_{tr}$ is the transmitted laser power. In the experiment, the transmitted laser power is 1.3± 0.1 µW, and the calculated contribution of the shot-noise to the rotation measurement is about $8.1\times 10^{-13}f\;\mathrm{rad}/s/\sqrt{\mathrm{Hz}}$. The noise-equivalent-power of the APD is about $2.75\;\mathrm{pW}/\sqrt{\mathrm{Hz}}$, and the sum of the two contributions is then $4.6\times 10^{-12}f\;\mathrm{rad}/s/\sqrt{\mathrm{Hz}}$. This is a kind of white phase noise, which increases with the Fourier frequency. In the frequency band below 1 Hz, these two noise contributions are negligible when compared to the other noise sources.

The two leak-out beams pass through different optical paths in the Mach–Zehnder beam combiner. Therefore, the environmental disturbances cause different phase variations of the two beams, thereby generating extra noise in the beat note. The length of the non-common beam path is about 75 cm. In order to measure the additional noise of this part, we built a similar Mach–Zehnder-type demonstration optical apparatus in order to simulate the perturbation effect on the beat note. In this experiment, a laser beam is divided in two, and an AOM is inserted in one of the beams as a frequency shifter with a modulation frequency of 75 MHz. After the beams are combined, the beat note is detected by an APD that is of the same type as the one used in the Sagnac frequency detection. The result is shown in Figure 6, where the dark yellow curve shows the beat frequency noise of the MZ interferometer. We find that in the higher frequency band of 5 Hz to 1 kHz, the performance is mainly affected by vibration and acoustic noise in the laboratory, while in the lower frequency band below 5 Hz, it is mainly affected by the air flow and the temperature fluctuation. The additional noise of the beam combiner has a non-negligible impact on the PRG in the region of 0.01 to 2 Hz. However, it is necessary for us to conduct further research and make some improvements, for example by the implementation of a monolithic beam combiner and better environmental isolation.

### 4.5. Scale Factor Fluctuation

The noise contribution of the scale factor fluctuation is contained in $S_{\nu ,sagnac}^{2}$ in Equations (Equation 8) and (Equation 9). The laser gyroscope converts the measurement of rotation rates into a measurement of frequency, where $K_{s}=\frac{4A}{\lambda P}\cos\theta$ is the oriented scale factor in units of Hz/rad·s. It depends on the laser wavelength λ, the enclosed area *A*, the perimeter *P* of the ring cavity, and the angle between the gyroscope area normal vector and the Earth rotation axis θ. Taking the derivative on both sides of Equation (Equation 1), we obtain:(13)$$\frac{\delta f_{s}}{f_{s}}=\frac{\delta K_{s}}{K_{s}}+\frac{\delta \Omega_{E}}{\Omega_{E}}.$$

Equation (Equation 13) indicates that the relative stability of the Sagnac frequency depends on the stability of the scale factor and the stability of the Earth rotation rate itself. In order to obtain EOPs from the PRG, the measurement resolution should reach $\delta \Omega_{E}/\Omega_{E}=10^{-10}$. Therefore, the required stability of the scale factor should even exceed $10^{-10}$. Since the orientated scale factor $K_{s}$ is a function of *A*, *P*, λ, and θ, we assume that the perturbation of $K_{s}$ is $\delta K_{s}$ and is given by:(14)$$\delta K_{s}=\frac{\partial K_{s}}{\partial A}\cdot \delta A+\frac{\partial K_{s}}{\partial \lambda }\cdot \delta \lambda +\frac{\partial K_{s}}{\partial P}\cdot \delta P+\frac{\partial K_{s}}{\partial \theta }\cdot \delta \theta .$$

Taking both sides of Equation (Equation 14) divided by $K_{s}$, we obtain:(15)$$\frac{\delta K_{s}}{K_{s}}=\frac{\delta A}{A}-\frac{\delta \lambda }{\lambda }-\frac{\delta P}{P}-\delta \theta \cdot \tan\theta .$$

The last term is related to the angle θ, which was clearly described in previous articles [4,35]. To measure the contribution from the tilt, we use a tiltmeter with a nano-radian resolution to monitor the orientation change of the gyroscope platform. The calculated noise contribution is shown in Figure 6, colored in dark cyan as orientation fluctuation, and indicates that it is well below the shot-noise limit. Next, we consider the geometrical scale factor S=4A/(λP). It is clear that:(16)$$\frac{\delta S}{S}=\frac{\delta A}{A}-\frac{\delta \lambda }{\lambda }-\frac{\delta P}{P}.$$

Assuming an ideal case that the cavity is a perfect square cavity, then the area can be expressed as $A=P^{2}/16$. It is always true that the perimeter of the ring cavity is an integer multiple of the laser wavelength when the laser satisfies the resonant conditions, which is P=mλ. Therefore, Equation (Equation 16) is rewritten as:(17)$$\frac{\delta S}{S}=\frac{P\delta P/8}{P^{2}/16}-\frac{\delta P/m}{P/m}-\frac{\delta P}{P}=0.$$

Equation (Equation 17) shows that under ideal conditions, the perturbations of the area, perimeter, and laser wavelength are canceled out. This requires the resonance to be maintained and the longitudinal mode index *m* to remain the same. This assumption holds if the cavity is a perfect square and the laser always resonates with the cavity, then S=4A/(λP)=m/4, and the changes on the cavity length and area are completely compensated by the accompanying change of the laser wavelength. That is, as long as the longitudinal mode index *m* does not change, the geometrical scale factor *S* would always remain the same in an ideal case, which is known as the self-compensation mechanism [20,35].

For a realistic situation, Reference [36] gave an excellent discussion about the geometrical stability for a non-rigid body cavity, where they decomposed the cavity deformations into six degrees of freedom (DOF): one diagonal common-mode stretching, one differential mode stretching, two shear planar deformations, one diagonal tilt, and one out-of-plane tilt. The conclusion is that the geometrical stability is affected by the diagonal common-mode stretching to the first order, while the other five DOFs only affect to the second order. However, if we take the self-compensation mechanism into account, the perturbations of the scale factor would depend on the diagonal common-mode stretching to the second order because the laser wavelength λ always changes with the perimeter *P*. Therefore, the perturbations of the scale factor rely on all six DOF deformations to the second or higher orders. Based on the measurement of the cavity length fluctuation in Section 4.2, we find that the scale factor fluctuation is currently well below the shot-noise limit of our PRG, which is too low to be depicted in Figure 6.

### 4.6. Torsional Signal Detection

We note that there are three distinct peaks near 20 Hz in the noise curve of our PRG colored in blue in Figure 6. In order to determine whether the peaks come from the torsional swing of the granite support platform of the gyroscope, we place two seismometers on and underneath the granite table and take the recordings simultaneously, as shown in Figure 7a. The results are depicted in Figure 7b, where the solid lines colored in red, green, and pink are the acceleration records on the granite table in the north-south (NS), east-west (EW), and vertical directions (Z), respectively. The dashed lines colored in orange, black, and dark cyan are the acceleration records on the lab floor in the NS, EW, and Z directions, respectively. It can be found from the collocated observations of the two seismometers that the resonant peak near 15.8 Hz is caused by a swing of the platform in the NS direction, and the resonant peak at 18.8 Hz results from a swing in the EW direction. The resonant peak near 24.5 Hz is excited by torsional motion. It can be proven by the comparison of the data between the seismometers and the PRG in Figure 7b, where the rotational noise curve of the PRG is colored in blue and indicated with the right axis. Since the torsional motion represents a rotation and the seismometer is primarily sensitive to translations, the amplitude of the torsional peak is about one order of magnitude weaker than the translational motion in the observations of the seismometer, as shown in Figure 7b. While the PRG is only sensitive to rotation, the amplitude of the torsional peaks around 24.5 Hz is larger compared to the other two peaks at 15.8 Hz and 18.8 Hz. The two translational peaks remain observable in our PRG because they contain rotational components in their signals. We find that the amplitude of the rotational components detectable by our laser gyroscope is less than 10nrad after integration, as shown in Figure 6 [15].

## 5. Conclusions

With the increasing demand for highly performing rotational sensors for applications in the geosciences and fundamental physics, the development of highly accurate large-scale laser gyroscopes is urgently needed, in which the PRGs will have an important role due to their unique features. The configuration of an empty cavity excited by two external laser sources not only has the advantage of higher storage power, but can also be helpful in the understanding of the systematics from the laser dynamics of their ring laser counterparts. The establishment of the absolute scale factor and all systematic errors are essential for the measurement of the EOPs, Lense–Thirring frame dragging detection, and moreover, the search for the Lorentz violation by such high-resolution gyroscopes [37].

We characterized the noise processes in a passive laser gyroscope and identified them from the derived noise transfer function of the PRG. We found that in different frequency ranges, the most prominent noise sources of the gyroscope are different. In the frequency range below $10^{-2}\;\mathrm{Hz}$, the contribution of the FSR is the dominant limitation, which is caused by the drift of the ring cavity length. In the band between $10^{-2}$ and $10^{3}\;\mathrm{Hz}$, the main limitation is the noise of the frequency discrimination system, which strongly depends on the frequency discrimination slope. Since the frequency discrimination slope acts as a low-pass filter, in the frequency bands above the bandwidth of the cavity, the noise contribution of the frequency discrimination system increases with the Fourier frequency. In the frequency region below 2 Hz, the RAM effect becomes the main factor. Furthermore, the additional noise introduced by the Mach–Zehnder-type beam combiner is a significant contributor in the 0.01–2 Hz frequency band. These three noise sources are currently the main limiting factors of our passive laser gyroscope.

The shot-noise limit of our gyroscope at present is $6\times 10^{-10}\;\mathrm{rad}/s/\sqrt{\mathrm{Hz}}$. In order to further improve the performance of the PRG, the noise of the frequency discrimination system needs to be well controlled, especially the noise caused by the RAM effect. We hope to suppress it to the shot-noise level in the frequency region of $10^{-3}$ to 1Hz. For the FSR jitter, an active control scheme of the cavity length is required with a goal for the cavity length fluctuation to drop below $10^{-11}\;m/\sqrt{\mathrm{Hz}}$ [31]. The additional noise contribution from the beam combiner must be reduced by a more compact design and better environmental isolation. Apart from that, the back-scattering noise, caused by the imperfect mirror surface and the noise introduced by the injected laser beam pointing jitter in the PRG, also need a quantitative analysis, which was not within the scope of this paper.

## Figures and Tables

**Figure 1 sensors-20-05369-f001:**
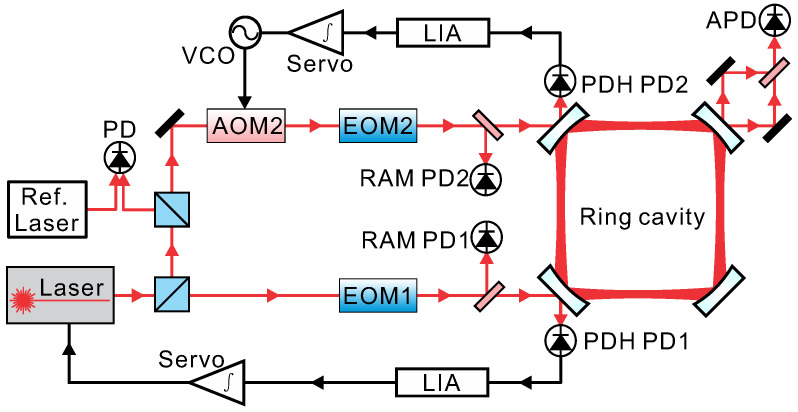
Experimental scheme of the PRG. AOM, acousto-optic modulator; EOM, electro-optic modulator; RAM PD, photo-diode for residual amplitude modulation (RAM) detection; PDH PD, photo-diode for PDH locking; LIA, lock-in amplifier; VCO, voltage controlled oscillator; APD, avalanche photo-diode; Ref. Laser, an ultra-stable laser as a reference to diagnose the ring cavity length fluctuation.

**Figure 2 sensors-20-05369-f002:**
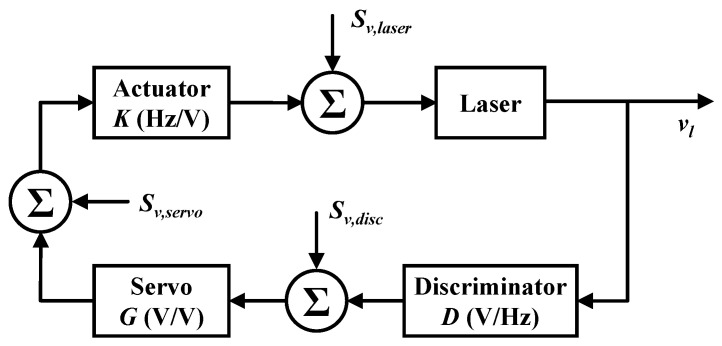
Block diagram of a laser frequency stabilization system. $S_{\nu ,laser}$, frequency noise of the free-running laser; $\nu_{l}$, laser output frequency; $S_{\nu ,disc}$, noise introduced by the frequency discrimination system; $S_{\nu ,servo}$, noise introduced by the servo amplifier [29,30].

**Figure 3 sensors-20-05369-f003:**
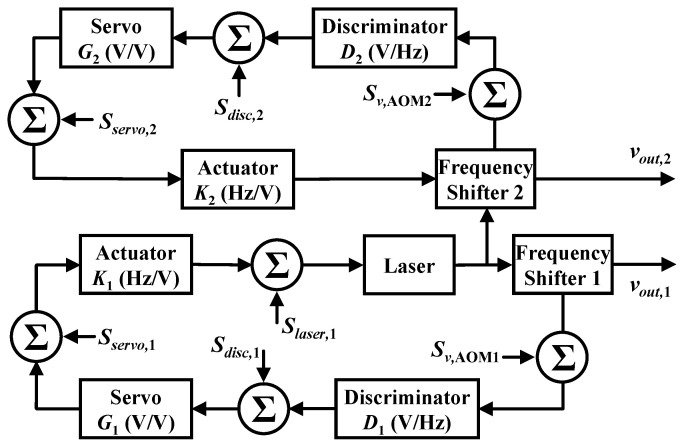
Block diagram of a PRG system.

**Figure 4 sensors-20-05369-f004:**
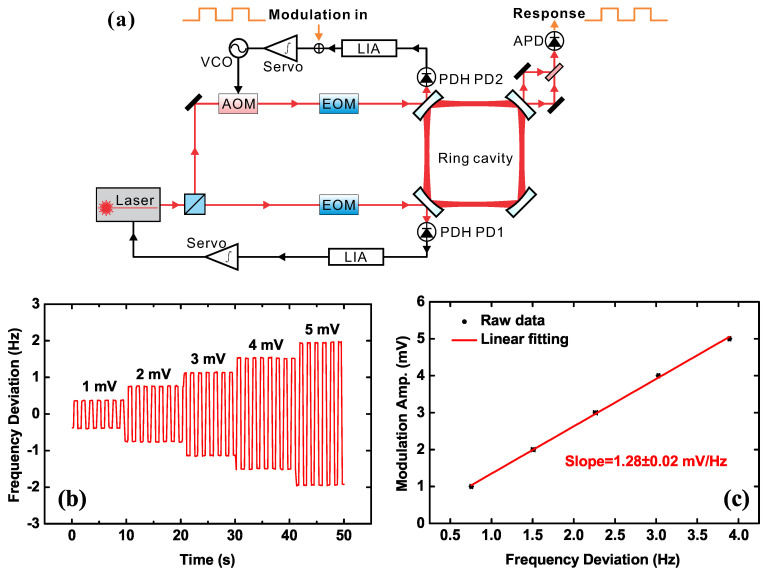
Modulation measurement of the discriminator slope. (**a**) The experimental setup; a modulation signal is fed into the PDH error signal of the secondary loop, and a corresponding signal appears in the output laser frequency. (**b**) Responses under different modulation amplitudes. (**c**) The measured data and a linear fitting of the slope.

**Figure 5 sensors-20-05369-f005:**
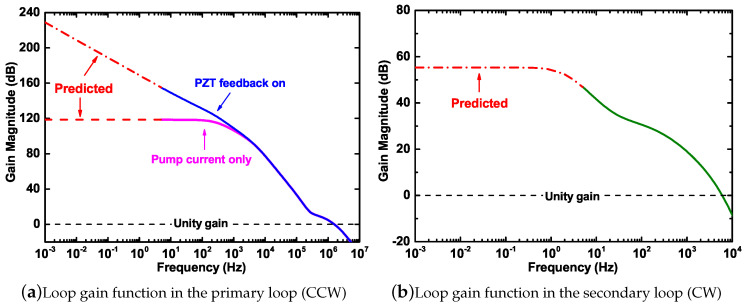
The loop gains of both locking loops. The solid lines represent the measured data, and the dashed lines are the predicted data, based on the poles-zeros settings of the PID regulators.

**Figure 6 sensors-20-05369-f006:**
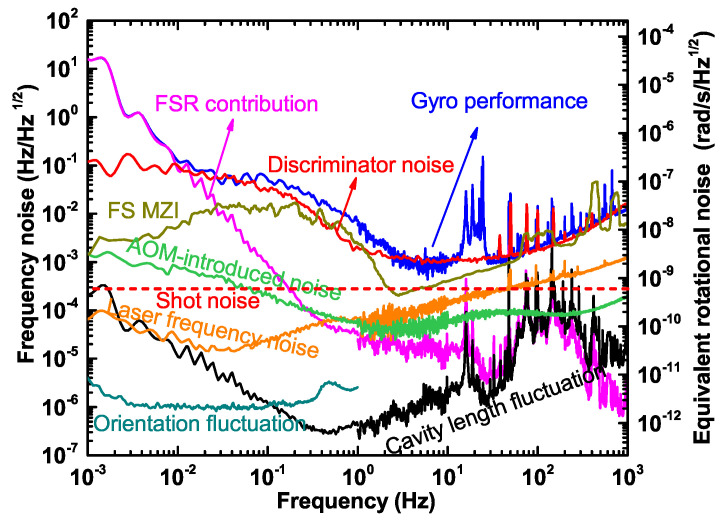
The measured performance of the PRG and the evaluated noise contributions. Blue line, the performance of the PRG; red solid line, the noise contribution of the discrimination system; dark yellow line, frequency noise of the free-space MZ interferometer (FS MZI); pink line, noise contribution of the FSR; green line, frequency noise introduced by AOM; orange line, frequency noise of the free-running laser; dark cyan, noise contribution of the orientation; black line, contribution of the residual cavity length fluctuation; red dashed line, shot-noise limit of the PRG.

**Figure 7 sensors-20-05369-f007:**
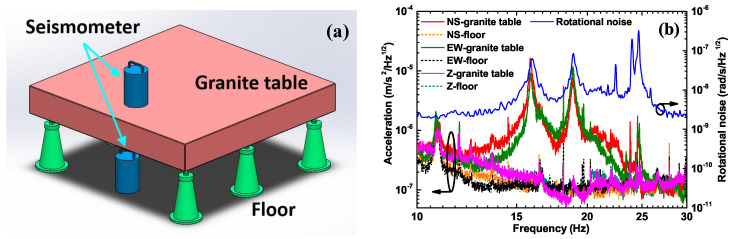
The torsional components’ measurement setup and results. (**a**) The differential acceleration measurement setup. (**b**) The recordings’ comparison. The solid lines colored in red, green, and pink are the acceleration records on the granite table in the north-south (NS), east-west (EW), and vertical directions (Z), respectively. The dashed lines colored in orange, black, and dark cyan are the acceleration records on the lab floor in the NS, EW, and Z directions, respectively. The blue curve is the rotational noise of the PRG for comparison, indicated with different vertical axes and arrows. It should be noted that there is no absolute linkage between the left axis and the right one.

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
