# Peer review of "Noise Analysis of a Passive Resonant Laser Gyroscope"

_sensors, 2020, doi:10.3390/s20185369_

Round 1

Reviewer 1 Report

COMMENTS

In this article, the authors investigate the limitations of a large-scale passive resonator laser gyro (PRG) in terms of the characterization of its main noise processes because these devices can play a fundamental role in geophysical measurements for the determination of the earth orientation parameters (in this case, with a precise determination of earth rotation rate, aiming to achieve a sensitivity equivalent to 10-10 parts to one). For this, they first derive the transfer function of the gyro and then, they identify all the noise sources and calculate analytically their respective contributions. All the analytic process and mathematical derivations seem correct. Then, based on an experimental design, they perform an estimation of the system parameters to obtain the measurement of the performance and the contribution of different noise sources as a function of frequency. Finally, as a result of the analysis of  the noise sources, they try to find design recommendations to reduce or minimize the value of these errors.

After having read the entire manuscript, it seems to me technically correct. On the other hand, taking into account other recent works published recently on this topic by the same authors (Ref. 15 and 19 of this manuscript), I think that the present work seems solid and robust enough because it was sufficiently tested in those previous publications. On the other hand, the experimental results obtained in this work seem logic and convincing to achieve the PRG performance improvement.

However, in my modest opinion, there is a lack of references to important works or projects which have been carried out on similar topic during last recent years (like LIGO project and others) that have been omitted here. The results of these works would be of high interest to add here. This is because large-scale interferometers have been used in these experiments and would result good adding their main measurement results in terms of accuracy and resolution.

On the other hand, while reading the entire manuscript, I have found several typographical errors that I wish to transmit to the authors to improve the final text of the manuscript. They are collected next:

  1. Line 18: replace text ''... at some latitude θ ...'', with ''... at some co-latitude θ ...''.

  1. Line 22: replace text ''... Ω the rotation rate of the gyro mounting frame ...'', with ''... Ω the earth rotation rate ...''.

  1. Line 37: replace text ''... sensitivity of 10-14-1 rad/s ...'', with ''... 10-14 rad/s . Hz -1/2 ...''.

Author Response

Response to Reviewer 1 Comments

Reviewer 1:

We thank the reviewer 1 for the encouraged comments.

COMMENTS AND SUGGESTIONS FOR AUTHORS

Point 1: In this article, the authors investigate the limitations of a large-scale passive resonator laser gyro (PRG) in terms of the characterization of its main noise processes because these devices can play a fundamental role in geophysical measurements for the determination of the earth orientation parameters (in this case, with a precise determination of earth rotation rate, aiming to achieve a sensitivity equivalent to 10-10 parts to one). For this, they first derive the transfer function of the gyro and then, they identify all the noise sources and calculate analytically their respective contributions. All the analytic process and mathematical derivations seem correct. Then, based on an experimental design, they perform an estimation of the system parameters to obtain the measurement of the performance and the contribution of different noise sources as a function of frequency. Finally, as a result of the analysis of the noise sources, they try to find design recommendations to reduce or minimize the value of these errors.

After having read the entire manuscript, it seems to me technically correct. On the other hand, taking into account other recent works published recently on this topic by the same authors (Ref. 15 and 19 of this manuscript), I think that the present work seems solid and robust enough because it was sufficiently tested in those previous publications. On the other hand, the experimental results obtained in this work seem logic and convincing to achieve the PRG performance improvement.

However, in my modest opinion, there is a lack of references to important works or projects which have been carried out on similar topic during last recent years (like LIGO project and others) that have been omitted here. The results of these works would be of high interest to add here. This is because large-scale interferometers have been used in these experiments and would result good adding their main measurement results in terms of accuracy and resolution.

Response 1: According to the suggestion, we add three references [1-3] related to the LIGO and Virgo experiments, and add explanations about large-scale Michelson interferometer and Sagnac interferometer at two different places in the paper.

[1] Abbott, B. P. et al. (LIGO Scientific Collaboration and Virgo Collaboration). Observation of Gravitational Waves from a Binary Black Hole Merger. Phys. Rev. Lett. 2016, 116, 061102.

[2] Abbott, B. P. et al. (LIGO Scientific Collaboration and Virgo Collaboration). Prospects for observing and localizing gravitational-wave transients with Advanced LIGO, Advanced Virgo and KAGRA. Living Rev. Relativ. 2018, 21, 3

[3] Acernese, F. et al. (Virgo Collaboration). Advanced Virgo Status. J. Phys.: Conf. Ser. 2020, 1342, 012010.

In line 15 to line 20 of the marked copy, we add a sentence “Optical interferometers have important applications in the field of precision measurement. Large-scale interferometers have much better sensitivity and resolution, therefore many ground-breaking works benefited from their high sensitivities have been reported. For example, large-scale Michelson interferometers, such as LIGO and VIRGO, make it possible to detect extremely weak gravitational-wave signals [1-3]. In the meantime, large-scale Sagnac interferometers play a vital role in measuring the rotation of the Earth [4].”.

In line 92 to line 93 of the marked copy, we add a sentence “Successful examples have been carried out in gravitational wave detection by using a passive cavity-enhanced sensing technology [1].”.

Point 2: On the other hand, while reading the entire manuscript, I have found several typographical errors that I wish to transmit to the authors to improve the final text of the manuscript. They are collected next:

Line 18: replace text ''... at some latitude θ ...'', with ''... at some co-latitude θ ...''.

Line 22: replace text ''... Ω the rotation rate of the gyro mounting frame ...'', with ''... Ω the earth rotation rate ...''.

Line 37: replace text ''... sensitivity of 10-14-1 rad/s ...'', with ''... 10-14 rad/s. Hz-1/2 ...''.

Response 2: In line 38 of the marked copy, we rewrite the sentence as “The detection signal for a horizontally placed geodetic laser gyroscope at some co-latitude  on the Earth is proportional to the rotation rate and can be expressed as [4]:”.

In line 40, we correct the sentence as “where  is the output frequency of the sensor (Sagnac frequency), is the geometrical scale factor, is the orientated scale factor,  is the area enclosed by the light path of the gyroscope,  is the perimeter of the ring cavity,  is the laser wavelength,  the Earth rotation rate, and  the north-south angle of projection.”.

In line 57, we correct the sentence as “Ring laser technology has kick-started the field of rotational seismology, where ultimately a rotation measurement resolution of 10-14-1 rad/s and frequency bandwidth of 3 mHz-50 Hz is demanded.” The mentioned data here should be resolution in units of rad/s.  

Reviewer 2 Report

The paper by Kui Liu et al. analysed the noise of a passive resonant laser gyroscope developed at HUST. The work is very important and useful for the development of the passive resonant laser gyroscope. It is well written. I recommended the acceptance. The only concern is the predicted curve in Fig. 5. It doesn’t mention how it obtains and also missed the discussion.

Author Response

Response to Reviewer 2 Comments

Reviewer 2:

We thank the reviewer 2 for the careful review and the encouraging judgment.

COMMENTS AND SUGGESTIONS FOR AUTHORS

Point 1: The paper by Kui Liu et al. analysed the noise of a passive resonant laser gyroscope developed at HUST. The work is very important and useful for the development of the passive resonant laser gyroscope. It is well written. I recommended the acceptance. The only concern is the predicted curve in Fig. 5. It doesn’t mention how it obtains and also missed the discussion.

Response 1: To explain the predicted curve in Figure 5, we add a sentence “where the predicted curves are calculated by the poles-zeros settings of the proportional-integral-derivative (PID) regulators” in lines 244 and 245, and corresponding modification is also made to the caption of Figure 5 as “…based on the poles-zeros settings of the PID regulators”.

As for the discussion of the predicted curve, we add a sentence in line 250 to line 251, “The predicted gain values are used to calculated the suppression ratios of the cavity length fluctuation and the laser frequency noise.”

To be clearer, in line 307, we change the sentence “The residual cavity length fluctuation noise is not a limitation at this moment, which benefits from the high primary loop gain.” into “The residual cavity length fluctuation noise is not a limitation at this moment, which benefits from the sufficient primary loop gain shown in Figure 5 (a).”

We also change the sentence in lines 326 and 327 “A broader bandwidth of the locking loop and higher loop gains are required to suppress the noise contributions further, which is essential to reach the shot-noise of this instrument.” into “Since the loop-gain of the secondary loop is only 55 dB in the frequency band below 1 Hz, as shown in Figure 5 (b), a broader bandwidth of the locking loop and higher loop gains are required to suppress the noise contributions further, which is essential to reach the shot-noise of this instrument.”

Reviewer 3 Report

I have the following notes:

  • Please, consider adding ORC ID
  • Line 4: Earth orientation parameters (EOP)
  • Line 16: You can mention the Sagnac effect usage in the general point of view. Important in Accurate time transmission.
  • Please, provide a general Introduction without equations or use equation after general Introduction.
  • Line 37: typo mistake
  • Please, add more relevant information about your purpose, not only what you did but did you obtain in comparison with appropriate references. Furthermore, I strongly suggest adding the State of Art Section into your article with RELEVANT and newer references. You have some more recent references, but most of them are quite old, which seems like the topic is "quite old".
  • Line 91: ... is shown in Figure 1, not Fig. 1. Please, use the same notation as is in a caption.
  • The usage of AOM may lead to lower the diffraction efficiency, did you take it into account?
  • Figure 2 is under copyright, please, add the references in the caption [17,18].
  • Equation 2 is also under copyright, and you did not mention reference for. However, you did a small notation "correction," but the equation was published here:
  • Line 330: Scale factor fluctuation
  • May you add more details about torsional vibration in your scheme/measurement?
  • Please, follow the template with reference style.

The overall impression of the article is positive, but the article has to be improved in the Introduction Section, and all questions have to be addressed in the article. I miss the future work paragraph. Please, provide more details about Sagnac effect in comparison with Sagnac interferometer.

I chose novelty “low” due to your not recent references, please, provide us more most recent references.

Author Response

Response to Reviewer 3 Comments

Reviewer 3

We are thankful to the comments of reviewer 3, which helps to improve the manuscript considerably.

COMMENTS AND SUGGESTIONS FOR AUTHORS

I have the following notes:

Point 1: Please, consider adding ORC ID

Response 1: The ORCIDs of the authors are:

Yuanbo Du: https://orcid.org/0000-0002-7912-9143

Karl Ulrich Schreiber: https://orcid.org/0000-0002-3775-5058

Zehuang Lu: https://orcid.org/0000-0002-2800-325X

Jie zhang: https://orcid.org/0000-0003-2885-8837

Point 2: Line 4: Earth orientation parameters (EOP)

Response 2: In line 4 and line 73, we change “Earth orientation parameters (EOPs)” into “Earth orientation parameters (EOP)”.

Point 3: Line 16: You can mention the Sagnac effect usage in the general point of view. Important in Accurate time transmission. Please, provide a general Introduction without equations or use equation after general Introduction.

Response 3: We rewrite the introduction part as the following:

“Optical interferometers have important applications in the field of precision measurement. Large-scale interferometers have much better sensitivity and resolution, therefore many ground-breaking works benefited from their high sensitivities have been reported. For example, large-scale Michelson interferometers, such as LIGO and VIRGO, make it possible to detect extremely weak gravitational-wave signals [1-3]. In the meantime, large-scale Sagnac interferometers play a vital role in measuring the rotation of the Earth [4].

Sagnac effect is useful in detention of rotational signals. Instruments based on Sagnac effect have many applications in different fields [4-16]. In particular, large-scale optical gyroscopes find applications in inertial navigation, geophysical study, seismic isolation and platform stabilization, etc [4, 6-9]. Optical gyroscopes utilize the non-reciprocal phenomenon inside a ring cavity introduced by the rotation of the cavity frame based on the Sagnac effect, which means two beams experience unequal round-trip travel time in opposite directions inside an identical light path of a rotating ring interferometer. The interferometric gyroscopes convert the travel time difference of the opposite beams into the accumulated phases difference, while the resonant gyroscopes measure the frequencies difference by utilizing a ring cavity instead. The state-of-the-art interferometric fiber optical gyroscopes can reach a sensitivity of 10-9 rad/s/√Hz [10,11]. Recently, micro-optical gyros made of whispering gallery mode resonators have attracted a lot of attentions and is believed to have great potential to be applied to both industrial and navigational fields [12,13]. On the other hand, a branch of resonant laser gyroscopes have evolved from compact aircraft inertial sensors back into large-scale and complex laboratory setups for the application in the geosciences in recent decades, with the sensitivity as high as 10-11 rad/s√Hz [4,7,14,15,16].”

Point 4: Line 37: typo mistake

Response 4: In line 57, we correct the sentence as “Ring laser technology has kick-started the field of rotational seismology, where ultimately a rotation measurement resolution of 10-14-1 rad/s and frequency bandwidth of 3 mHz-50 Hz is demanded.”

Point 5: Please, add more relevant information about your purpose, not only what you did but did you obtain in comparison with appropriate references. Furthermore, I strongly suggest adding the State of Art Section into your article with RELEVANT and newer references. You have some more recent references, but most of them are quite old, which seems like the topic is "quite old".

Response 5: In line 64, we change the sentence “Another application is in the field of geodesy.” into “Our aim is to apply it to the field of geodesy.” to make it clearer about our purpose.

We add 11 new citations in our manuscript as the following:

[1] Abbott, B. P. et al. (LIGO Scientific Collaboration and Virgo Collaboration). Observation of Gravitational Waves from a Binary Black Hole Merger. Phys. Rev. Lett. 2016, 116, 061102.

[2] Abbott, B. P. et al. (LIGO Scientific Collaboration and Virgo Collaboration). Prospects for observing and localizing gravitational-wave transients with Advanced LIGO, Advanced Virgo and KAGRA. Living Rev. Relativ. 2018, 21, 3.

[3] Acernese, F. et al. (Virgo Collaboration). Advanced Virgo Status. J. Phys.: Conf. Ser. 2020, 1342, 012010.

[5] Exertier, P.; Samain, E.; Bonnefond, P.; Guillemot, P. Status of the T2L2/Jason2 experiment. Adv. Space Res. 2010, 46, 1559-1565.

[6] Passaro, V. M. N.; Cuccovillo, A.; Vaiani, L.; De Carlo, M.; Campanella, C. E. Gyroscope Technology and Applications: A Review in the Industrial Perspective. Sensors 2017, 17, 2284.

[8] Martynov, D.; Brown, N.; Nolasco-Martinez, E.; Evans, M. Passive optical gyroscope with double homodyne readout. Opt. Lett. 2019, 44, 1584-1587.

[9] Ren, W.; Luo, Y.; He, Q.; Zhou, X.; Deng, C.; Mao, Y.; Ren, G. Stabilization Control of Electro-Optical Tracking System with Fiber-Optic Gyroscope Based on Modified Smith Predictor Control Scheme. IEEE Sens. J. 2018, 18, 8172-8178.

[10] Li, Y.; Cao, Y.; He, D.; Wu, Y.; Chen, F.; Peng, C.; Li, Z. Thermal phase noise in giant interferometric fiber optic gyroscopes. Opt. Express 2019, 27, 14121-14132.

[11] Guattari, F.; Bernauer, F.; Laudat, T.; Ponceau, D. Rotational GroundMotion Instrumentation: Characterization and Improvements. AGU Fall Meeting 2019, 2019, S21G-0594.

[12] Liang, W.; Ilchenko, V. S.; Savchenkov, A. A.; Dale, E.; Eliyahu, D.; Matsko, A. B.; Maleki, L. Resonant microphotonic gyroscope. Optica 2017, 4, 114-117.

[13] Lai, Y. H.; Suh, M. G.; Lu, Y. K.; Shen, B.; Yang, Q. F.;Wang, H.; Li, J.; Lee, S. H.; Yang, K. Y.; Vahala, K. Earth rotation measured by a chip-scale ring laser gyroscope. Nat. Photonics 2020, 14, 345-349.

Point 6: Line 91: ... is shown in Figure 1, not Fig. 1. Please, use the same notation as is in a caption.

Response 6: We change the notation into “Figure” in the context and all figure captions throughout the manuscript.

Point 7: The usage of AOM may lead to lower the diffraction efficiency, did you take it into account?

Response 7: In our passive laser gyroscope, the diffraction efficiency of the AOM is not a problem because the laser power is sufficient to be injected in. And the AOM is also a core element for feedback locking and power stabilization of certain loops.

Point 8: Figure 2 is under copyright, please, add the references in the caption [17,18].

Response 8: Figure 2 is modified according to Ref. [29,30], so we add the references in the caption of Figure 2. We also rewrite the sentence in line 152 as “From the perspective of control theory, a typical block diagram of the feedback control loop in a laser frequency stabilization system is given by T. Day et al. and shown in Figure 2.”

Point 9: Equation 2 is also under copyright, and you did not mention reference for. However, you did a small notation "correction," but the equation was published here:

Response 9: Equation 2 is original from Ref. [29], and we have already given the reference in the paper, as shown in line 162 of the summited version. The cited paper is the same as the one mentioned by the reviewer.

Point 10: Line 330: Scale factor fluctuation

Response 10: We correct “scale factor fluctuation” into “Scale factor fluctuation” in line 358.

Point 11: May you add more details about torsional vibration in your scheme/measurement?

Response 11: We rewrite Sec 4.6 from line 398 to line 418 as the following.

 “We note that there are three distinct peaks near 20 Hz in the noise curve of our PRG colored in blue in Figure 6. In order to determine whether the peaks come from the torsional swing of the granite support platform of the gyroscope, we place two seismometers on and underneath the granite table and take the recordings simultaneously, as shown in Figure 7 (a). The results are depicted in Figure 7 (b), where the solid lines colored in red, green, and pink are the acceleration records on the granite table in north-south (NS), east-west (EW), and vertical directions (Z), respectively. The dashed lines colored in orange, black, and dark cyan are the acceleration records on the lab floor in NS, EW, and Z directions, respectively. It can be found from the collocated observations of the two seismometers that the resonant peak near 15.8 Hz is caused by a swing of the platform in the NS direction and the resonant peak at 18.8 Hz results from a swing in the EW direction. The resonant peaks near 24.5 Hz is excited by torsional motion. It can be proved by the comparison of the data between the seismometers and the PRG in Figure 7(b), where the rotational noise curve of the PRG is colored in blue and indicated with the right axis. Since the torsional motion represents a rotation and the seismometer is primarily sensitive to translations, the amplitude of the torsional peak is about one order of magnitude weaker than the translational motion in the observations of the seismometer, as shown in Figure 7 (b). While the PRG is only sensitive to rotation, the amplitude of the torsional peaks around 24.5 Hz is larger compared to the other two peaks at 15.8 Hz and 18.8 Hz in the observations of the PRG. The two translational peaks remain observable in our PRG because they contain rotational components in their signals. We find that the amplitude of the rotational components detectable by our laser gyroscope is less than 10 nrad after integration, as shown in Figure 6 [15].”.

Point 12: Please, follow the template with reference style.

Response 12: We change the reference style of the manuscript according to the template, as shown in lines 494, 496, 498, 506, 510, 521, 523, 524, 526, 528, 541.

Point 13: The overall impression of the article is positive, but the article has to be improved in the Introduction Section, and all questions have to be addressed in the article. I miss the future work paragraph. Please, provide more details about Sagnac effect in comparison with Sagnac interferometer.

Response 13: We rewrite the introduction part, and please see the reply of question 3. We discuss the future work in the last paragraph, from line 440 to line 448. The discussion about Sagnac effect and Sagnac interferometer is included in the new introduction part.

Point 14: I chose novelty “low” due to your not recent references, please, provide us more most recent references.

Response 14: We add 11 new references. The list is summarized in the reply of question 5.

Round 2

Reviewer 3 Report

Dear authors,

the authors improved their paper according to the comments. 

I have the last and minor comment. Please, consider a replacement of "HUST" in the title. Nobody knows what does it mean from the abbreviation. 

Is it necessary to have “ Developed at HUST“ in the title? If yes, please, consider the full text instead of the abbreviation.

Thank you for your interesting paper, and I wish you all the best in your research.

Author Response

Reviewer 3

We are thankful to the useful comment of reviewer 3, which helps to improve the manuscript considerably.

COMMENTS AND SUGGESTIONS FOR AUTHORS

Dear authors,

the authors improved their paper according to the comments.

Point 1:  I have the last and minor comment. Please, consider a replacement of "HUST" in the title. Nobody knows what does it mean from the abbreviation.

Is it necessary to have “Developed at HUST” in the title? If yes, please, consider the full text instead of the abbreviation.

Thank you for your interesting paper, and I wish you all the best in your research.

Response 1: We delete “Developed at HUST” in the title.

In the abstract line 5 and line 6 in the marked copy, we make a  modification of the sentence “In order to extend the relative rotation measurement accuracy to a better level so that it can be used for the determination of the Earth orientation parameters (EOP), we investigate the limitations in passive resonant laser gyroscopes (PRGs) to pave the way for future development.” into “In order to extend the relative rotation measurement accuracy to a better level so that it can be used for the determination of the Earth orientation parameters (EOP), we investigate the limitations in a passive resonant laser gyroscope (PRG) developed at Huazhong University of Science and Technology (HUST) to pave the way for future development.”

We also change the sentence in line 96 and line 97 “In this paper, we take a full investigation of the noise sources in a PRG with a size of 1 m×1 m that is developed at Huazhong University of Science and Technology (HUST) [15].” into “In this paper, we take a full investigation of the noise sources in a PRG with a size of 1 m×1 m developed at Huazhong University of Science and Technology (HUST) [15].”.
